# Assessment of Encapsulated Probiotic *Lactococcus lactis* A12 Viability Using an In Vitro Digestion Model for Tilapia

**DOI:** 10.3390/ani14131981

**Published:** 2024-07-05

**Authors:** Marcelo Fernando Valle Vargas, María Ximena Quintanilla-Carvajal, Luisa Villamil-Diaz, Ruth Yolanda Ruiz Pardo, Francisco Javier Moyano

**Affiliations:** 1Grupo de Investigación en Procesos Agroindustriales (GIPA), Doctorado en Biociencias, Facultad de Ingeniería, Universidad de La Sabana, Campus del Puente del Común, Km. 7, Autopista Norte de Bogotá, 250001 Chía, Cundinamarca, Colombia; marcelovalva@unisabana.edu.co (M.F.V.V.); maria.quintanilla1@unisabana.edu.co (M.X.Q.-C.); luisa.villamil@unisabana.edu.co (L.V.-D.); ruth.ruiz@unisabana.edu.co (R.Y.R.P.); 2Departamento de Biología y Geología, Universidad de Almería, 04120 Almería, Spain

**Keywords:** probiotic, *Lactococcus lactis*, encapsulated, in vitro, digestion, *Nile tilapia*

## Abstract

**Simple Summary:**

During their transit through the gastrointestinal tract (GIT) of fish, probiotics could be affected by a reduction in their viability. Therefore, it is important to evaluate how in vitro GIT conditions affect viable cells since probiotics must reach the intestine in adequate amounts to confer host health benefits. We found that GIT conditions such as stomach pH, residence time, and enzyme quantity affected the viability of probiotic bacteria. The evaluation of the influence of GIT conditions on probiotic viability using in vitro models is a useful tool for improving probiotic protection and feed supplementation.

**Abstract:**

Probiotics face harsh conditions during their transit through the gastrointestinal tract (GIT) of fish because of low-pH environments and intestine fluid. Therefore, the evaluation of probiotic viability under simulated gastrointestinal conditions is an important step to consider for probiotic supplementation in fish feed prior to in vivo trials. Therefore, this study aimed to evaluate the effect of stomach and intestinal simulated conditions on the viability of encapsulated *Lactococcus lactis* A12 using an in vitro digestion model for tilapia. A Box Behnken design was used to evaluate the potential effect of three factors, namely stomach pH, residence time in the stomach, and enzyme quantity, on the viability of encapsulated *Lactococcus lactis* A12. As the main results, low pH (4.00), long residence time (4 h), and enzyme quantity (2.68 U of total protease activity) led to lower final cell counts after the phases of the stomach and intestine. Encapsulated probiotic bacteria showed higher viability (*p* < 0.05) and antibacterial activity (*p* < 0.05) against the pathogen *Streptococcus agalactiae* than non-encapsulated bacteria. The results suggest that *L. lactis* A12 survives in GIT conditions and that the proposed in vitro model could be used to explore the viability of probiotic bacteria intended for fish feed supplementation.

## 1. Introduction

In 2024, global aquatic animal production was estimated at 185 million tons, which is a 4.0% increase from 2020. For the first time, aquaculture production exceeded capture, representing 51% and 49%, respectively. Of all aquaculture production (94 million tons), 84% was inland aquaculture, with tilapia and carp being the main species harvested [1]. The growing demand for aquatic food, particularly fish species, has intensified fish farming, leading to high stocking densities, degraded water quality, and the emergence of human-induced stressors that can make fish more susceptible to infections, ultimately reducing their growth performance [2,3,4].

As a response to managing these infections, fish farmers have used antibiotics to treat bacterial diseases; however, this approach has determined the appearance of antimicrobial-resistant microorganisms [5]. For this reason, environmentally friendly alternatives have been used to control pathologies, such as probiotics [4]. Probiotics are viable microorganisms that confer health benefits to a host when ingested in sufficient amounts [6]. The use of probiotics in aquaculture has been widely reported, and several papers describe its benefits including improvement in weight gain, immunomodulation, nutrient digestibility, and resistance to pathogens [7,8,9]. Lactic acid bacteria and Bacilli species are the most common microorganisms used in aquaculture [10,11]. Among lactic acid bacteria, the *Lactococcus* genus, particularly *Lactococcus lactis*, stands out as a promising probiotic candidate due to its generally recognized safety (GRAS) [12]. The species *L. lactis* A12 demonstrated probiotic characteristics including non-hemolytic behavior, resistance to simulated gastric conditions, and antibacterial activity to pathogens such as *Streptococcus agalactiae* and *Aeromonas hydrophila* [13]. In addition, in vivo trials have reported that dietary administration of *L. lactis* to *Nile tilapia* (*Oreochromis niloticus*) improved growth performance, intestinal microbiota, morphology, immune response, and pathogen resistance [14,15,16]. In particular, the strain *L. lactis* A12 used in this study improved growth performance, gut histology, gut microbiota, immune regulation, and resistance to infection of *Nile tilapia* fingerlings in an in vivo trial [17].

Probiotics are administered directly to the fish in the water or included as a feed additive. Addition to feed can be performed by mixing probiotics with feed or by spraying a probiotic solution on feed [18]. Regardless of the method used to supplement feed, probiotics must face harsh conditions in the fish gastrointestinal tract (GIT), such as an acidic environment, bile salts, and gastric and intestinal digestive enzymes [19,20,21,22]. These conditions can reduce the concentration of viable cells, leading to probiotic bacteria not reaching the intestine in enough amounts to exert their beneficial effects [23].

Since probiotic viability can be affected by the conditions of the GIT, protective techniques such as encapsulation must be used [24]. Although encapsulation can protect probiotics from GIT conditions, the evaluation of probiotic viability under simulated digestive conditions using in vitro systems is a useful tool to consider before the determination of a supplementation dose [25]. These systems have been used for different purposes, that is, to evaluate feed digestibility for poultry [25] and aquatic animals [26,27], as well as evaluation of probiotic viability for humans [28,29] and several fish species [21,22,23,30]. Systems simulating in vitro conditions present in the digestive tract of humans and animal species are valuable tools that are regularly used to evaluate this loss to obtain practical information that may help design better vehiculation systems or more suitable doses [28].

However, the design and operating conditions of in vitro simulations must be closely based on the physiology of the species to obtain valuable results that can be correlated with the in vivo response. In addition, in vitro models could help minimize the cost of in vivo trials and the number of fish needed [9]. In this sense, most studies aimed at evaluating the tolerance of probiotics to human GIT conditions consider gastric and intestinal phases separately and not as a sequential process [19,20,28,29,31]. Furthermore, in studies that simulate fish digestion, factors involved in hydrolysis such as pH, temperature, type and amount of enzymes, and residence time in the stomach and intestine are frequently not based on the real physiological conditions present in live fish and are different from those reported by several authors in in vivo trials [26,32,33,34].

In relation to this, it is important to note that the combination of standardized in vitro assays with some statistical procedures, such as response surface methodology (RSM), is a powerful tool that allows modeling the effects of the in vitro digestion process under a combination of different GIT conditions [26,35]. Taking this into account, the present study aimed to evaluate the effect of various conditions that can affect the viability of probiotic *L. lactis* A12 during digestion using an in vitro system adapted to simulate the GIT of tilapia.

Therefore, this study aimed to evaluate different conditions in the stomach and intestine of an in vitro digestion process using the response surface methodology tool to explore the viability of encapsulated probiotic bacteria intended for feed supplementation in fish nutrition.

## 2. Materials and Methods

### 2.1. Ethical Statement

The project followed the Colombian national government’s regulations. A permit for accessing genetic resources was issued by the Colombian Ministry of Environment Number 117 (Otrosí4) on the 8 May 2018 for five years.

### 2.2. Microorganisms

*L. lactis* A12, *Priestia megaterium* M4, and *Priestia* sp. M10 were isolated from a competitive exclusion bacterial culture derived from the gut microbiota of *Nile tilapia* gut microbiota in a previous study by Melo-Bolivar et al. [36]. Then, Melo-Bolivar et al. [13] identified these bacteria using molecular techniques, sequenced the whole genome and evaluated their probiotic potential, including antibacterial activity against *Streptococcus agalactiae*, susceptibility to antibiotics, hemolytic activity, hydrophobicity, and survival in response to pH and bile salts. The bacteria were deposited under codes A12 (*L. lactis* A12), M4-MR4 (*Priestia megaterium* M4), and M10-MR10 (*Priestia* sp. M10) in the Chilean Collection of Microbial Genetic Resources (CChRGM) at the Instituto de Investigaciones Agropecuarias (INIA, Chillan, Chile). This institute is registered in the World Data Centre for Microorganisms (WDCM) with registration number 1067.

### 2.3. Preparation of Culture Medium and Fermentation Conditions

The methodology for the preparation of the culture medium and the fermentation procedure is described in the previous work of Valle-Vargas et al. [37]. The bioreactor conditions were the agitation speed (150 RPM), temperature (28 °C), and incubation time (17 h). Finally, after the process had finished, a culture medium with grown probiotic bacteria was stored at 4 °C for further use.

The culture composition and the speed of agitation in the bioreactor were achieved in previous experiments through design optimization (manuscript under submission).

### 2.4. Encapsulation of Probiotic Bacteria

#### 2.4.1. Preparation of Feed Solution

The feed solution was prepared by mixing 600 mL of a 17 h incubation culture medium containing probiotic bacteria with maltodextrin (MD) and whey powder (WH) at a final mass fraction of 43.4% solids. The MD/WH mass ratio was 0.25:0.75. The final mixture was homogenized using a EURO-ST 20 HS laboratory stirrer (IKA^®^, Wilmington, NC, USA) at 1600 rpm for 2 min. This final mixture was the feed solution.

#### 2.4.2. Spray Drying of Probiotic Bacteria

A pilot-scale spray dryer (GEA Process Engineering Mobile MinorTM, GEA Niro, Dusseldorf, Germany) equipped with a pneumatic nozzle (1 mm diameter) was used to produce spray-dried probiotics. The feed solution was constantly homogenized using a magnetic stirrer while being pumped with a peristaltic pump (MARLOW 520S, WATSON, Falmouth, UK) at a rate of 50 mL/min. The atomizing air pressure was set at 1.4 bar. The input and outlet temperatures were set at 180 and 90 °C, respectively. The spray-dried encapsulated probiotic (ENCP) was collected in a stainless-steel container, packed in aluminum bags under vacuum, and stored at 4 °C for further use. Non-encapsulated probiotic (N-ENCP) was produced by spray drying, and a culture medium was obtained without the addition of wall materials for comparison purposes under optimal conditions.

The whey/maltodextrin proportion and air pressure were achieved in previous experiments through the design optimization of the spray drying process (manuscript under preparation).

#### 2.4.3. Viability of Probiotic Spray-Dried Powder

One gram of spray-dried probiotic was added to 9 mL of phosphate buffer solution (PBS), and then, 10-fold serial dilutions were made. Viable cell counting was performed by plate counting using the drop plate method, in which 20 µL of each dilution was dropped onto the surface of TSA and allowed to dry. Finally, the agar plates were incubated at 28 °C for 24 h. After 24 h, colonies were counted in each plate, and the final cell concentration was expressed as log_10_ CFU/g [38].

It is important to note that the viability of the *Priestia* species was not evaluated by in vitro digestion for two reasons: (1) their cell counts were too low to detect using the drop plate method, and (2) our main interest was in evaluating the viability of *L. lactis* A12 due to its probiotic potential.

### 2.5. In Vitro Assays

For the assessment of probiotic viability in an in vitro digestion system that simulates the GIT of tilapia, the experimental approach described by Gilannejad et al. [26] was used with some modifications. The present study consisted of two steps: (1) a set of experiments one at a time (OFAT) and (2) RSM design.

The initial experiments involved a preliminary evaluation of the effect of stomach pH, digestion time, and probiotic dosage on the viability of probiotic bacteria. The second was a Box Behnken design that was used to assess the combined effect of some of the previously evaluated factors on the survival of probiotic bacteria included in a feed matrix.

#### 2.5.1. Preparation of Tilapia Enzymatic Extract

The enzyme extracts required for the assays were obtained from adult tilapia (*O. niloticus*) (N = 30; average mass 352 ± 15 g) provided by SPAROS (Portugal). Fish were fed ad libitum with a single meal (commercial feed) in the morning and were sampled 4 h after feeding to ensure a maximum amount of digestive enzymes in the intestine. Fish were anesthetized and then killed with a 2-phenoxyethanol overdose and were immediately frozen at −20 °C prior to being sent to the laboratory at the University of Almería (Spain) where they were dissected to separate the stomach and the digestive tract. Manipulation of fish was performed in accordance with the Guidelines from the European Union Council (2010/63/EU). Extracts required were prepared by mechanical homogenization of the intestinal portion in distilled water (1:10 *w*/*v*) followed by centrifugation (3220× *g*, 20 min, 4 °C). The supernatant was then filtered through a dialysis system with an MWCO of 10 kDa (Pellicon XL, Millipore^®^, St. Louis, MO, USA), and the concentrated extracts were freeze-dried until being used in the assays. Total alkaline protease activity was measured according to Kunitz’s method [39] modified by Walter [40] using casein as a substrate. The assays were carried out using a set of small glass vials (25 mL) as bioreactors in which enzymes and substrates were mixed and maintained under continuous and gentle agitation using multiple stirrers. The complete arrangement was placed within an incubation chamber at a fixed temperature of 28 °C. Furthermore, the activity of trypsin and chymotrypsin was determined according to the methodology described by Uscanga et al. [34]. Total protease and specific protease activity were expressed as enzymatic units per gram of freeze-dried enzyme extract (U/g). Finally, total amylase activity was measured at pH 7.5, following the 3,5-di-nitrosalicylic acid (DNS) method [41], using starch as a substrate. Amylase activity was expressed as enzymatic units per gram of freeze-dried enzyme extract (U/g).

#### 2.5.2. First Set of Assays: One Factor at a Time

Stomach pH, stomach residence time, and probiotic dosage were selected as key factors that could influence probiotic viability [28,42]. The justification for the selection was as follows:

Stomach pH: The selected range of pH values was based on those measured in *Nile tilapia* stomachs according to [33,43]. For the assay, one gram of probiotic was suspended in 4 mL of sterile distilled water adjusted at pH 4.0, 5.0, or 6.0 using HCl 1.0 M, and the solutions were incubated at 28 °C for 4 h.

Stomach residence time: Residence times between 2 and 6 h were tested based on gut transit times reported for the species [32,33]. For the assay, one gram of probiotic was suspended in 4 mL of sterile distilled water, and pH was adjusted at the lower pH tested (4.0) in the previous experiment using HCl 1.0 M. The incubation was maintained for 2, 4, and 6 h at 28 °C.

Probiotic dosage: In this experiment, different initial doses of probiotics between 75 mg and 1000 mg were faced with simulated digestion that included both the stomach and intestinal stages. For the assay, the variables mentioned above of probiotics were suspended in 4 mL of sterile distilled water, and the pH was adjusted at 5.50 using HCl 1.0 M, followed by incubation for 2 h at 28 °C to simulate stomach conditions. Subsequently, the pH was raised to 7.40 using NaOH 1M, followed by mixing with 4 mL of PBS (containing 45 µM sodium taurocholate, 10 mM CaCl_2,_ and 2.13 U total protease activity). This amount was calculated to simulate a 100 g fish, considering the average amount of total protease related to fish weight measured in the fish used for the preparation of enzyme extracts (see Section 2.4.1.) The incubation of this intestinal stage was maintained for an additional period of 2 h. Samples were taken before and after the stomach and intestinal phase for cell counts.

Samples were taken before and after stomach pH, stomach residence time, and probiotic dosage assays for the determination of the cell count. First, 100 µL of a sample was added to 900 µL of PBS. Then, serial 10-fold dilutions were made. The drop plate method described in Section 2.4.3 was used to determine cell count.

#### 2.5.3. Second Set of Assays: RSM Design

For this set of assays, the probiotic was included in a simplified feed matrix to more closely reproduce the expected effects and interactions that could take place within the GIT of the fish. The preparation of the feed matrix was carried out according to the methodology described by Sribounoy et al. [20] with some modifications. Fish meal (11%), corn meal (28%), soybean meal (39%), and wheat meal (22%) were used as feed ingredients. The dry meals were ground, sieved (425 µm), and sterilized at 121 °C for 15 min. Then, they were mixed, and the probiotic powder (ENCP) was added to the mixture to obtain an initial concentration of 5.99 ± 0.09 log_10_ CFU/g feed. Sterile distilled water was added to obtain a homogeneous dough that was used for the in vitro assays.

The experiment was developed using a Box Behnken design (BBD) combining the factors that had a significant effect on the viability of the probiotic in the previous set of assays. These were stomach pH (4.00–6.00), stomach residence time (1–4h), and intestinal enzyme quantity (1.58–2.68 U expressed as total protease activity per gram of feed). The fixed factors in all the assays were the pH and total length of the intestinal phase (7.4 and 5 h, respectively). The design matrix consisted of 15 runs with 3 replicates at the central point (Table 1). In vitro assays were developed following a procedure similar to that described in the last part of the previous section but varying the conditions according to the BBD matrix. Cell counts were determined at the initial and final time for the stomach, intestine, and total process using the drop plate count method described in Section 2.4.3. The bacterial change in the stomach (SC), intestine (IC), and total (TC) was expressed as logarithmic bacterial change (log_10_ CFU) between the initial and final cell counts (log_10_ CFU/mL) and established as the response variable.

#### 2.5.4. Model Validation

The models for the response variables (SC, IC, and TC) were expressed as coded equations as a function of the independent factors. Coded equations are useful to identify the relative impact of factors by comparing factor coefficients (coefficient estimates).

Model validation was carried out by comparing the predicted and experimental values of the response variables under the most favorable and harsh conditions for the encapsulated probiotic. The percentage of error between the predicted and experimental data was calculated. Validation runs were performed in triplicate.

#### 2.5.5. Antibacterial Activity (AA) of In Vitro Digestion of Probiotic-Supplemented Feed

The conditions selected from the model validation were selected to evaluate the antibacterial activity of the final in vitro digestion product of feed supplemented with encapsulated probiotic (ENCP) and non-encapsulated probiotic (N-ENCP) against the pathogen *S. agalactiae*. Twenty-one hours of in vitro digestion was carried out: 4 h in the stomach and 17 h in the intestine. AA was determined by the methodology described by Valle-Vargas et al. [37] and was expressed as an inhibition zone (mm). In vitro digestions were performed in triplicate.

### 2.6. Statistical Analysis

For the OFAT approach, for stomach pH, stomach residence time, and probiotic dosage, an ANOVA test was performed with a level of significance of 0.05. The LSD test was used to determine the difference between treatments with a confidence level of 95%. ANOVA assumptions such as normal distribution, homogeneity of variance, and independence were verified for OFAT and BBD data. The BBD was built, and data were analyzed using the Design Expert software version 11.0.0 (Stat-ease Inc., Minneapolis, MN, USA).

## 3. Results

### 3.1. OFAT Experiments

The encapsulated probiotic bacteria had a viable cell count of *L. lactis* A12 and *Priestia* species of 7.76 ± 0.19 and 3.59 ± 0.06 log_10_ CFU/g, respectively. The effect of pH on the viability of encapsulated *L. lactis* A12 is presented in Figure 1. Significant bacterial reduction (*p* < 0.05) was observed at the lowest pH tested, accounting for 8.06% (0.54 log_10_) of the control value measured before incubation. In contrast, the viability measured at pH 6.0 and 7.0 increased significantly by 0.60 and 0.51 log_10_ CFU, respectively.

The effect of the total incubation time at pH 4.0 on the count of viable cells of *L. lactis* is shown in Figure 2. Negative correlation was evident since, compared to the initial count (6.71 log_10_ CFU/mL), the number decreased by 5.81, 8.19, and 11.17% at 2, 4, and 6 h, respectively.

The effect of the initial dose of probiotics on the bacterial counts after incubation in the stomach and intestinal phase is presented in Figure 3 and Figure 4.

Independently of the dose used, the final cell count after the simulated stomach incubation was significantly higher (*p* < 0.05) than the initial (Figure 3). Bacterial counts increased by 0.26, 0.30, 0.44, 0.20, 0.14, and 0.14 log_10_ CFU for dosages of 75, 100, 250, 500, 750, and 1000 mg, respectively. The final viable cell count was higher for the 1000 mg dose, followed by 750, 500 mg, and 250 mg. Finally, the 75 and 100 mg probiotic doses presented the lowest final cell count for the stomach phase and did not show a significant difference (*p* > 0.05) between them.

A similar increasing trend was observed for the intestinal phase (Figure 4), except for the 250 mg dose. In this case, *L. lactis* viable cells increased by 0.24, 0.21, 0.20, 0.39.0.39, and 0.57 log units for 75, 100, 250, 500, 750, and 1000 mg, respectively. The minimum initial dosage that allowed a cell concentration of 6.00 log_10_ CFU/mL to be obtained after the intestine phase was 250 mg (6.12 log_10_ CFU/mL).

### 3.2. Box Behnken Design (BBD) of In Vitro Assays

The results obtained in the BBD are presented in Table 1. Data from encapsulated *L. lactis* A12 viable cell count changes after stomach digestion were fitted to a linear model, while those corresponding to intestinal and whole digestion data were fitted to a quadratic model (Table 2). The lack of fit was not significant for the total change data (*p* > 0.05). No significant terms were considered for the generated models. The models of stomach, intestinal, and total digestion explained 86.65%, 92.51%, and 99.07% of the total variability presented in the experiment.

The ANOVA test shows that for the stomach stage, the terms pH (A) and digestion time (B) had a significant effect (*p* < 0.05) on the bacterial change of *L. lactis* A12. For the intestinal stage and total digestion, individual terms (A, B and C) and quadratic terms (A^2^ and B^2^) significantly (*p* < 0.05) influenced viable cells of probiotic bacteria. A significant (*p* < 0.05) interaction was obtained between stomach pH and residence time (AB) only for total digestion.

It can be observed that for the SC model, stomach pH has a major impact on probiotic viability compared to residence time; however, the latter had a negative impact. For IC and TC, terms A and B had a positive effect on the response; however, the enzyme (C), quadratic terms (A^2^ and B^2^), and interaction (AB) had a negative influence on bacterial change.
SC (log_10_ CFU) = −0.65 + 0.69[A] − 0.21[B] − 0.04[C] (1)
IC (log_10_ CFU) = 1.66 + 0.36[A] + 0.48[B] − 0.27[C] − 0.45[A^2^] − 0.59[B^2^](2)
TC (log_10_ CFU) = 1.09 + 1.02[A] + 0.33[B] − 0.31[C] − 0.17[AB] − 0.42[A^2^] − 0.54[B^2^](3)

In general, except for a few conditions observed in the BBD matrix, the increase in bacterial counts during the intestinal stage (0.14 to 1.84 log_10_ CFU) compensated for their reduction in the stomach stage. Figure 5A–C present contour plots indicating the intensity of the combined effects evaluated for each digestion stage on the viability of *L. lactis* A12. It can be observed that the highest bacterial reduction occurred at a low pH (4.0), while a higher pH (6.00) led to positive bacterial changes (from −1.60 to 0.35 log_10_ CFU). Increasing residence time in the stomach at low pH also negatively affected bacterial changes (Figure 5A).

Intestine bacterial change values ranged from −0.14 to 1.84 log_10_ CFU. Positive bacterial changes were achieved at high stomach pH and residence time (Figure 5B). Meanwhile, the quantity of enzymes had a negative effect. Higher IC values were observed at a low enzyme level (1.58 U/g feed) (Figure 5C). However, most bacterial changes were positive regardless of the quantity of enzymes.

Finally, the total bacterial changes in GIT ranged from −1.52 to 1.82 log_10_ CFU (Figure 5D,E). This response presented the same trend as IC. It was possible to observe that encapsulated probiotics that faced harsh conditions such as low pH (4.00) and longer residence times (4 h) in the stomach showed a lower and higher reduction in viable cell counts in the intestine and total process.

### 3.3. Validation of the Model

Two conditions were used to validate the SC, IC, and TC models (Table 3). The experimental error between the predicted and observed values was lower than 10%. The initial cell count of *L. lactis* A12 was 5.00 log_10_ CFU/mL. After in vitro digestion, *L. lactis* A12 cell count decreased by 15.20% in run 1 and increased by 17.2% in run 2.

### 3.4. Antibacterial Activity of In Vitro Digestion

The final cell counts and antibacterial activity of the probiotics ENCP and N-ENCP after in vitro digestion are shown in Table 4. It was observed that probiotic ENCP under a stomach pH of 5.50 showed a higher AA (*p* < 0.05) than probiotic N-ENCP. Additionally, the final cell count was significantly higher (*p* < 0.05) for probiotic ENCP compared to N-ENCP.

## 4. Discussion

The results of OFAT experiments evidenced that low pH and a high residence time in the stomach negatively affected bacterial viability. This could be an attribute of probiotic bacteria being susceptible to acid environments due to the presence of hydrogen ions (H^+^) in gastric solutions that can enter bacterial cells, decrease the cytoplasmic pH, and cause cell death [28,44]. Bacteria can withstand acidic stress through pH homeostasis which regulates pH inside and outside the cells by maintaining a high internal pH at low external pH; however, once a certain acidic condition is reached, the cytoplasmatic pH decreases, and pH homeostasis is destroyed. This causes damage to proteins and DNA, resulting in cell death [45]. Other studies have reported that low-pH conditions and long residence time affected the survival of encapsulated *Lactobacillus plantarum* [19] and *L. lactis* Gh1 [46].

Finally, the results of the probiotic dosage indicated that the probiotic bacteria at different concentrations of viable cells were not negatively affected by the presence of intestinal enzymes and less extreme pH [33,43]. Although the probiotic bacteria survived in intestinal conditions, a minimum number of viable cells is required to ensure that probiotics may exert their beneficial effects at the intestinal level (6.00 log_10_ CFU/g of feed) [20].

The results of the in vitro digestion evaluation of the viability of encapsulated *L. lactis* A12 included in a fish feed matrix showed that pH, residence time in the stomach, and enzyme levels influenced viability in the intestinal tract as well as the total change after digestion. In the intestine, the results suggested that the viability of probiotic bacteria was affected by previous conditions of pH and residence time in the stomach, and to a lesser extent by the amount of intestine enzymes. A possible explanation could be that the exposure of probiotic bacteria to acidic conditions and long residence time in the stomach negatively affects their further survival in the intestine, resulting in a reduction in viable counts at the end of digestion. It has been reported that pH changes taking place when passing from the stomach to the intestine could cause membrane damage and alteration of the structure of proteins and DNA in probiotic bacteria [47]. On the other hand, the increase in the relative concentration of intestinal digestive enzymes resulted in a reduction in bacterial growth. The presence of different enzymes (mostly proteases and lipases) could affect probiotic survival either due to their ability to hydrolyze the whey matrix used for encapsulation [48,49], making the bacteria more vulnerable to intestinal conditions [23], or by direct disruption of the cell membrane and DNA damage [44].

Regardless of the effect of the factors evaluated and except in a few conditions, five hours in the intestine was time enough to compensate for the loss of bacteria in the stomach. This could be related to the presence of bacteria in the intestinal tract in an alkaline environment (pH > 7.00), which is more favorable for their growth, as was observed in the OFAT experiments. In our study, the feed was supplemented at a concentration of 6.00 log_10_ CFU/g, which was enough to reach the intestine phase at the same concentration in in vitro digestion. In vivo trials in *Nile tilapia* found that probiotic feed supplementation with a dose between 10^4^ and 10^7^ CFU/g of feed improved growth performance, immunological parameters, microbiota modulation, and resistance against pathogens [50,51].

Concerning model validation, the models used had R^2^ values higher than 0.7. Also, the predicted R^2^ and adjusted R^2^ were in reasonable agreement since their values had a difference of less than 0.2. Adequate precision is another statistical parameter that measures the signal-to-noise ratio; our values were higher than 4.0, which means the models can be used to navigate design space. Even though the lack of fit for stomach and intestine change models was significant (*p* < 0.05), the experimental errors between predicted and observed values for the models under the two conditions evaluated were less than 10%. This means that the proposed models could be used to explore different GIT conditions in a sequential in vitro model that simulates digestive conditions for tilapia and to predict viability changes of probiotic bacteria.

In relation to the 21 h in vitro digestion of N-ENCP and ENCP probiotics, the latter showed a higher cell count. In several studies, encapsulation techniques have improved the survival of probiotic bacteria under GIT conditions, since wall materials protect bacteria against harsh conditions such as low pH, bile salts, and intestinal enzymes, exhibiting higher cell counts compared to non-encapsulated bacteria [21,22].

Probiotics exhibit antibacterial activity against pathogens through the production of bacteriocins, competitive exclusion, blocking of adhesion sites, and consumption of nutrients, among other mechanisms [52]. In the present study, probiotic bacteria (ENCP and N-ENCP) exhibited antibacterial activity against the fish pathogen *S. agalactiae* after in vitro digestion. Bacteria have been reported to produce bacteriocin and bacteriocin-like peptides in the presence of some microorganisms [53]. Melo-Bolivar et al. [17] reported that the probiotic consortium composed of *L. lactis* A12, *Priestia megaterium* M4, and *Priestia* sp. M10 exhibited antibacterial activity against *S. agalactiae* in vitro and in in vivo trials in fingerlings of *Nile tilapia*. In addition, other studies have reported that administration of encapsulated compared to non-encapsulated probiotics resulted in a higher survival rate after challenge with the fish pathogen *S. agalactiae* in in vivo trials in tilapia [23,54].

The use of encapsulation techniques reduced the loss of probiotics due to their protective effect [19]. In the present study, whey and maltodextrin were used as protective wall materials for *L. lactis* A12. These wall materials have been used for the encapsulation of *Lactobacillus fermentum* K73 by spray drying [55]. The authors reported that this mixture made probiotic bacteria more resistant to GIT conditions, mainly due to the protective effects of denatured whey proteins [28]. In addition, the use of spray drying results in the formation of a solid particle that represents a physical barrier that hinders the diffusion of compounds such as HCl into the capsule [55]. Reductions in bacterial counts obtained in the present study under low pH were close to those reported in the literature by several authors [20,21,22,55].

Similar approaches for evaluating probiotic viability using in vitro systems have been reported; however, many of these studies presented some limitations, such as the independent simulation of the stomach and intestine [19,20], the use of incubation conditions (temperature, pH, residence times) far from those really present in live fish [21,22,54], or the use of mammalian bile salts rather than enzymes extracted from fish [20,23]. This last point is particularly critical since the type and functionality of digestive enzymes secreted by fish are clearly different from their equivalents in mammals [42], and therefore, the results obtained in the simulations using the latter would not easily correlate with the response in live fish. In this sense, is important to develop physiologically based in vitro tests that consider different aspects like the dependence of digestive enzyme production on environmental and nutritional factors [34], gut transit times [32], and the total production of enzymes related to fish size [34,35].

## 5. Conclusions

It was possible to develop a sequential in vitro digestion model that mimics the GIT conditions of tilapia. It can be concluded that pH and residence time during the stomach stage are the main factors affecting probiotic viability and determining probiotic viability in the intestine. However, our results suggested that a high dosage (>6.00 log_10_ CFU/g of feed) of probiotics is not necessary for the probiotics to survive the passage through GIT and be able to exhibit antagonistic activity against *S. agalactiae*. Additionally, encapsulation of probiotic bacteria improved their viability and antagonistic activity after digestion.

The proposed model could be used to explore and evaluate probiotic viability under different conditions, allowing the improvement of probiotic protection and optimal dosing required for feed supplementation.

Finally, the proposed model could be adapted to other species of fish considering their particular physiological conditions.

## Figures and Tables

**Figure 1 animals-14-01981-f001:**
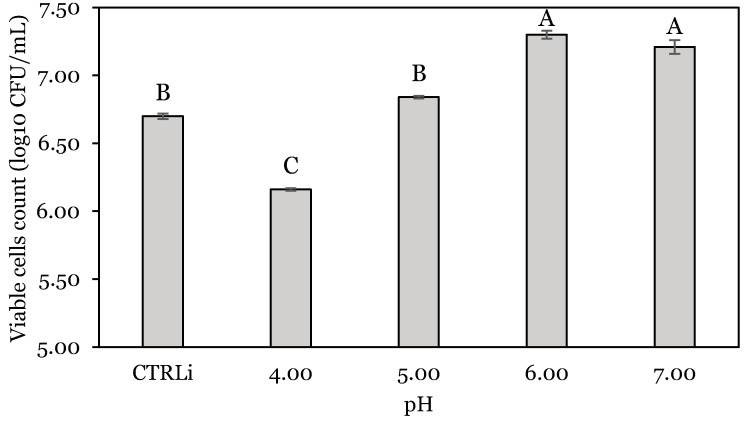
Viable cell counts of *L. lactis* A12 at different pH values. Different capital letters (A–C) indicate significant difference (*p* < 0.05). CTRLi: Initial cell count of *L. lactis* A12. Data are expressed as mean ± standard deviation.

**Figure 2 animals-14-01981-f002:**
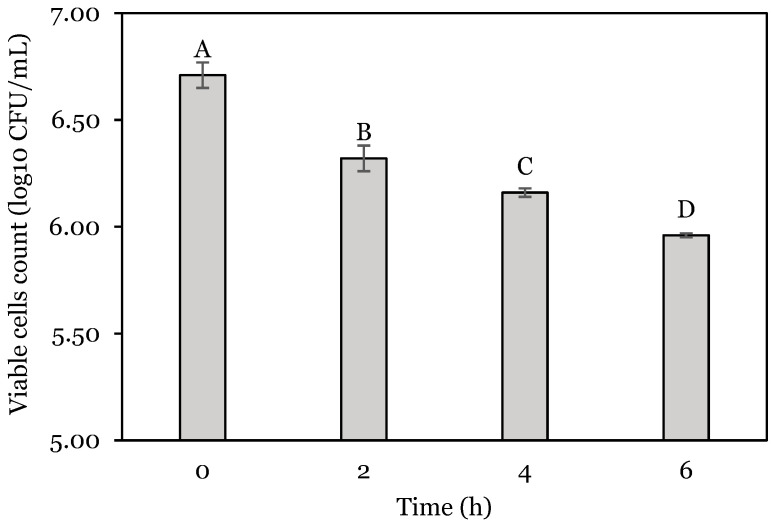
Viable cell counts of *L. lactis A12* at different times. Different capital letters (A–D) indicate a significant difference (*p* < 0.05). Data are expressed as mean ± standard deviation.

**Figure 3 animals-14-01981-f003:**
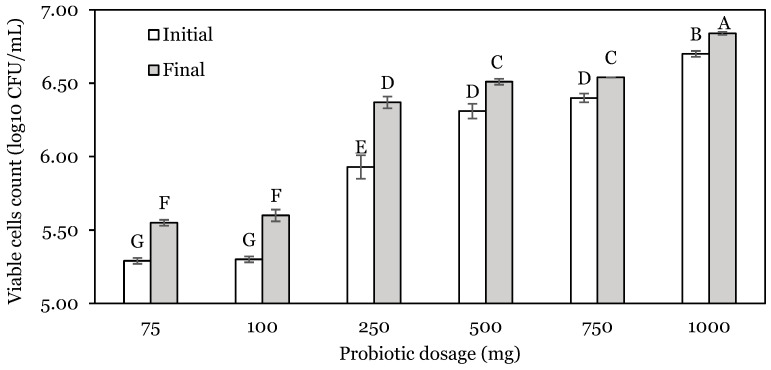
Viable cell counts of *L. lactis* A12 at different initial probiotic dosages for the stomach phase. Different capital letters (A–G) indicate significant differences (*p* < 0.05). Data are expressed as mean ± standard deviation.

**Figure 4 animals-14-01981-f004:**
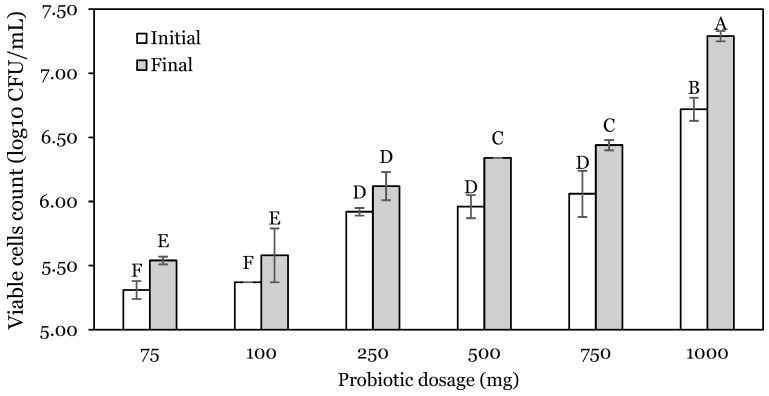
Viable cell counts of *L. lactis* A12 at different initial probiotic dosages for the intestine phase. Different capital letters (A–F) indicate a significant difference (*p* < 0.05). Data are expressed as mean ± standard deviation.

**Figure 5 animals-14-01981-f005:**
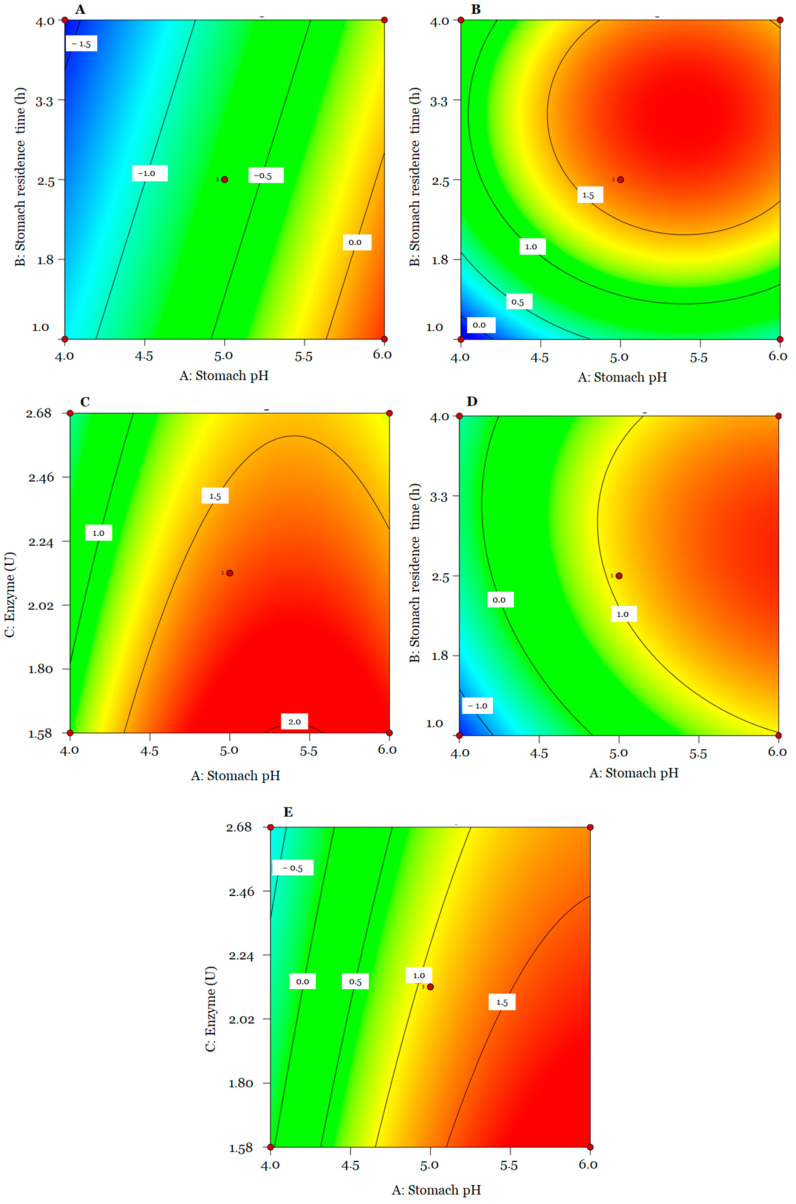
Contour plots for SC (**A**), IC (**B**,**C**), and TC (**D**,**E**).

**Table 1 animals-14-01981-t001:** BBD matrix with experimental results.

Run	Stomach: pH	Stomach Time: h	Enzyme: U	Stomach Change (SC): log_10_ CFU	Intestine Change (IC): log_10_ CFU	Total Change (TC): log_10_ CFU
1	5.00	1.0	2.68	−0.21	0.14	−0.06
2	6.00	2.5	1.58	−0.11	1.76	1.82
3	4.00	1.0	2.13	−1.51	−0.14	−1.52
4	5.00	4.0	2.68	−1.20	1.30	0.52
5	4.00	4.0	2.13	−1.60	1.06	−0.47
6	6.00	4.0	2.13	0.15	1.27	1.36
7	5.00	2.5	2.13	−0.72	1.74	1.01
8	5.00	1.0	1.58	−0.30	0.82	0.54
9	5.00	2.5	2.13	−0.75	1.75	1.11
10	5.00	4.0	1.58	−0.75	1.68	1.26
11	5.00	2.5	2.13	−0.78	1.84	1.05
12	6.00	2.5	2.68	−0.19	1.48	1.45
13	4.00	2.5	2.68	−1.07	0.19	−0.69
14	6.00	1.0	2.13	0.35	0.60	1.01
15	4.00	2.5	1.58	−1.17	1.07	0.14

Response variables results are expressed in logarithmic units (log_10_ CFU) for each combination of factor levels.

**Table 2 animals-14-01981-t002:** ANOVA and fitting parameters of in vitro assays.

	*p* Value		
Term	Stomach Change	Intestine Change	Total Change
Model	<0.0001	<0.0001	<0.0001
A: Stomach pH	<0.0001	0.0012	<0.0001
B: Stomach time	0.0289	0.0002	<0.0001
C: Enzyme	0.6314	0.0065	<0.0001
AB	-	-	0.0179
A^2^	-	0.0034	0.0001
B^2^	-	0.0006	<0.0001
Lack of fit	0.0123	0.0466	0.1307
R^2^	0.8665	0.9251	0.9907
R^2^ adjusted	0.8301	0.8836	0.9838
R^2^ predicted	0.7198	0.7705	0.9584
Adeq precision	14.4673	14.8264	42.4524

A *p*-value lower than 0.05 indicates a significant model term.

**Table 3 animals-14-01981-t003:** Validation of selected conditions.

	Factors			Predicted			Observed		
Run	pH	Time	Enzyme	SC	IC	TC	SC	IC	TC
1	4.00	4.0 h	2.68 U	−1.61	−0.46	−0.71	−1.55 (3.72%)	0.50 (8.69%)	−0.76 (7.04%)
2	5.50	4.0 h	2.68 U	−0.57	1.35	0.88	−0.54 (5.26%)	1.44 (6.66%)	0.89 (1.13%)

Values inside parentheses indicate experimental error (%) between predicted and observed response. Observed values are expressed as means.

**Table 4 animals-14-01981-t004:** Antibacterial activity of probiotics after in vitro digestion.

	Final Cell Count (log_10_ CFU/mL)	AA (mm)
N-ENCP_pH 4.00	7.40 ± 0.08 ^b^	5.16 ± 0.90 ^b^
N-ENCP_pH 5.50	7.49 ± 0.23 ^b^	4.99 ± 0.28 ^b^
ENCP_pH 4.00	7.91 ± 0.03 ^a^	5.34 ± 0.92 ^b^
ENCP_pH 5.50	7.92 ± 0.01 ^a^	5.95 ± 0.94 ^a^

Different lowercase letters (^a,b^) within the same column indicate a significant difference (*p* < 0.05). Data are expressed as mean ± standard deviation.

## Data Availability

The original contributions presented in this study are included in the article. Further inquiries can be directed to the corresponding author.

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
