# Peer review of "Assessment of Encapsulated Probiotic Lactococcus lactis A12 Viability Using an In Vitro Digestion Model for Tilapia"

_animals, 2024, doi:10.3390/ani14131981_

Round 1

Reviewer 1 Report

Comments and Suggestions for Authors

This manuscript titled "Assessment of encapsulated probiotic Lactococcus lactis A12 viability using an in vitro digestion model for tilapia" (Animals 3059616) established in vitro digestion model for evaluating probiotic viability in the GIT of tilapia. Results from in vitro experiment highlight the feasibility of this model to determine the actual supplementation and protective effect of probiotic additive in feed and provide the technical support for developing and applying new, effective, and sustainable probiotics in aquaculture.

Its main contents are valuable for the detection and application of probiotics as additive in aquafeed. But several parts of the main text should be streamlined, and this manuscript should be carefully proofread by a native speaker/professional editor to improve its whole readability and avoid any language mistakes. 

Major comments:

1. Regarding the "1.Introduction" section, multiple statements were overly long and redundant, and may confuse the readers. For example, in Line 39-44 and 53-56, these sentences are unnecessarily long and hard to follow. The descriptions (Line 70-72 and Line 75-80) on encapsulation and in vitro digestion system are correct, but seem to be a bit superfluous and wordy. It would be preferable to remove redundant/ unnecessary contents and rephrase the relevant statements for better understanding background in this study. 

2. Several descriptions in the section of "2.Materials and Methods" are lengthy and seem unnecessary. For example, the method descriptions in Line 201-203, Line 209-212, and Line 217-220 are redundant. Additionally, multiple sentences are quite long and complex in "2.Materials and Methods". For example, Line 255-258. The authors should compress or delete some descriptions to avoid long convoluted sentences and improve the whole readability.

3. In the section of "3.Resutls", there were some redundancies and long-windedness in the descriptions of results. Also, several statements in Line 275-284 are not concise and seem to be the description of the result analogy. The sentences in Line 340-342 can be transferred into the method section. It is suggested that the authors shorten the redundant contents or move those corresponding sentence to the discussion section for a clear and brief summary of main results in this study.

4. The main text of discussion part contains the repeating and long statements. Please streamline and organize the section of "Discussion" with emphasis for better clarifying your findings in this study. 

5. Check the references carefully according to the instructions for authors. Many errors are present in the current reference list, with many DOI missing, along with other inconsistencies like abbreviated vs. full journal names, italicized vs. non-italic latin name of species. For example, in Reference 5, DOI information should be provided. In Reference 3, 14, and 15, the latin name (scientific name) of species in the title should be indicated in italics. There are many same errors in the reference list.

Additionally, the unpublished manuscripts in the reference list should be removed. For example, Reference 44-45. Please check and modify accordingly.

Minor comments:

1. In Line 27, what about the enzyme quantity? Please specify the final result of enzyme quantity in the section of "Abstract".

2. Many journals currently limit the maximum number of author-provided keywords, with often no more 6 keywords. There were 7 keywords included in the text (Line 33). Moreover, some keywords in the current version are inaccurate. For examples, in Line 33, "fish" can be replaced with "Nile tilapia" or "Oreochromis niloticus". Please revise the "Keywords" section to meet the restriction on the number of keywords in this journal according to the information to related guides.

3. In "2.5.3. Second set of assays: RSM design" (Line 236-240), the formulation and proximal composition of experimental feed containing probiotic powder are not available in this study. The authors should give the relevant information in the corresponding part of the revised paper.

4. Table 1 and Table 2 lack the legends. It is suggested that the authors provide the table legends below the corresponding tables for more clearly showing results.

Other mistakes (highlighted in yellow) are presented in the PDF file.

Therefore, this manuscript will be reconsidered after major revision.

Comments on the Quality of English Language

The core content of this paper (Animals 3059616) under a title "Assessment of encapsulated probiotic Lactococcus lactis A12 viability using an in vitro digestion model for tilapia" is important for the development and application of probiotics in aquaculture and aquafeed.

However, there are several mistakes in the main text, particularly the confusing format of the reference list. Moreover, multiple redundant descriptions in the sections of "1.Introduction" and "4.Discussion" should be streamlined or directly removed for better clarity. Thus, it is strongly recommended that the whole paper should be polished by a native speaker to avoid any syntax or grammar errors in the revised version.

Reviewer 2 Report

Comments and Suggestions for Authors

The work is very good, with a robust methodology and results. It is not just more of the same work on probiotics, but presents an innovative and relatively simple methodology to solve problems faced by modern aquaculture. 

In the introduction, the authors could take advantage of the new SOFIA FAO and update the data.

in the methodology, does the description of the spraying of the probiotic have any references, if so, please cite them

describe which commercial feed was used and what pellet size and protein quantity

Reviewer 3 Report

Comments and Suggestions for Authors

Round 2

Reviewer 1 Report

Comments and Suggestions for Authors

This revised paper (Animals 3059616) entitled "Assessment of encapsulated probiotic Lactococcus lactis A12 viability using an in vitro digestion model for tilapia" has been carefully modified in response to the reviewers’ comment. As for the unchanged contents in the main text of the revised version, the authors have given the corresponding explanation in the list of response.

But there are still some minor errors in the reference list of this revised manuscript. For example, in reference 3 and reference 14, the scientific name of fish species in article title should be italicized. Regarding the reference 39, its title is written with capital letters for all words and is inconsistent with those of other references. Moreover, the full journal name is listed for reference 39, while other references use the abbreviated journal names. DOI information is missing in reference 5. Reference 44, 46, and 47 have no page number. The authors should check and revise the list of cited literature once more.

Although the current revision is suitable for acceptance, it still needs text editing to avoid minor errors.

Comments on the Quality of English Language

This revised paper (Animals 3059616) entitled "Assessment of encapsulated probiotic Lactococcus lactis A12 viability using an in vitro digestion model for tilapia" has been carefully modified in response to the reviewers’ comment. As for the unchanged contents in the main text of the revised version, the authors have given the corresponding explanation in the list of response.

But there are still some minor errors in the reference list of this revised manuscript. For example, in reference 3 and reference 14, the scientific name of fish species in article title should be italicized. Regarding the reference 39, its title is written with capital letters for all words and is inconsistent with those of other references. Moreover, the full journal name is listed for reference 39, while other references use the abbreviated journal names. DOI information is missing in reference 5Reference 44, 46, and 47 have no page number. The authors should check and revise the list of cited literature once more.

Although the current revision is suitable for acceptance, it still needs text editing to avoid minor errors.

Reviewer 3 Report

Comments and Suggestions for Authors

Dear Authors,

Thank you for your responses. The manuscript is much improved and ready for publication. However, I still have a small comment I would like to make as follows:

1- Lines 156, 231, 244 please use log10 CFU instead of Log10

2- Please check oC or o C. It must be consistent (Line 154, 177, 212, 229,...)

3- Line 259, delete a space before "AA was...".

4- Please make sure the authors are going to use (p<0.05, see line 291) or (p<0.05) or (p <0.05, see Line 327, 373, 374 and others)
